# New Models for Calculating the Maximum Compressive Force of Paper in Its Plane

**DOI:** 10.3390/ma16134544

**Published:** 2023-06-23

**Authors:** Paweł Pełczyński, Włodzimierz Szewczyk, Maria Bieńkowska, Zbigniew Kołakowski

**Affiliations:** 1Centre of Papermaking and Printing, Lodz University of Technology, Wólczańska 221, 95-003 Łódź, Poland; wlodzimierz.szewczyk@p.lodz.pl (W.S.); maria.bienkowska@p.lodz.pl (M.B.); 2Department of Strength of Materials, Lodz University of Technology, Stefanowskiego 1/15, 90-537 Łódź, Poland; zbigniew.kolakowski@p.lodz.pl

**Keywords:** maximum compressive force, models for load capacity, mechanical properties of paper

## Abstract

The main objective of the presented research was to find a model that describes the maximum compressive force of paper in its plane. The research began with crushing tests of a number of packaging paper samples of various lengths. It was shown that due to the specific structure of the paper and the high heterogeneity of its structure, packaging paper is material where it is difficult to determine the maximum compressive stress. Next, three analytical models describing the load capacity of a flat paper web were investigated and an alternative empirical model was proposed. The results of the performed tests are directly applicable in the calculation of the mechanical properties of corrugated cardboard and the determination of the load capacity of cardboard packaging.

## 1. Introduction

Predicting the load capacity of flat paper plates is of great practical importance due to the possibility of analysing the mechanical properties of corrugated board and packaging made of it, in particular, bulk packaging stored in high stacks [1,2,3,4,5,6,7,8,9,10,11,12]. Achieving high strength while keeping the weight of the packaging low leads to significant economic benefits. It is also crucial for minimizing the consumption of natural resources and the impact of the paper industry on the environment. Therefore, it is very important to design packaging with the best possible strength properties, which depend both on the geometric parameters of the corrugated cardboard used for its production and on the properties of the papers used [13,14,15]. The behaviour of the covering layers of cardboard is determined not only by the crushing resistance of the paper, but also by its geometric parameters (Wang et al. [16]), in particular the wavelength defining the distance between the places where the liner and flute (layer in cardboard) are glued together. With increasing this distance, the load-bearing capacity of the flat layer and the resistance of the cardboard to bending moments decreases significantly. Therefore, it is particularly important to be able to calculate the load capacity of the paper plate between the places where the liner layer is glued together with the flute [17,18,19]. Due to its complex internal structure and its high heterogeneity, paper is a material difficult to characterise and model (Rzepa and Hämäläine et al. [20,21]). The currently intensively developed measurement techniques based on the simultaneous analysis of universal testing machine data and the sequences of images of the tested samples are helpful here (Pełczyński et al. [22]). The authors also developed a technique based on the fusion of data from these two sources (Pełczyński et al. [23]), which facilitated further research work. The subject of the presented research was to develop an accurate, and at the same time easy to apply, practical model describing the load capacity of the paper web depending on its fastening length.

The ability to predict the load capacity of paper is of great importance in the task of determining the load capacity of corrugated board, honeycomb board and packaging produced from it, both by analytical methods and with the use of numerical techniques [24,25,26,27,28,29,30,31,32,33,34].

In the paper industry, one of the fundamental tests for paper properties is short-span compression test (SCT). It is performed on paper webs with a length of *l* = 0.7 mm. This test is conducted in the same way as for cylindrical metal samples under compression. The compressed cylindrical metal samples have a height-to-diameter ratio of 1.5.

In the production of cardboard packaging, it is also necessary to estimate the compressive strength of composite papers for longer spans. The testing machine used allowed for web lengths up to 5.0 mm. In [23], the authors presented a methodology for testing compressed paper webs, and this article is a continuation of that work, presenting a method for determining the load-bearing capacity of paper webs. The article also provides a formula for calculating the deflection mechanism without providing specific details about the mechanism itself and determining the load-bearing capacity. The authors are not aware of any of the international literature in which the load-bearing capacity of a paper web-based on the method of limit load (i.e., the method of state) is defined.

The unconventional organization of the text of the article results from a logical sequence of activities aimed at developing a model of the load-bearing capacity of flat paper webs. The next section describes known models for predicting the bearing capacity of flat plates that have been applied to paper. Then, the results of the measurements of the load capacity of selected packaging papers and the use of the proposed models for its prediction were presented. Section 4 contains an analysis of the obtained results, and Section 5 is devoted to the analysis of the applicability range of the proposed models. Section 6 shows the development of an alternative empirical model characterised by a good fit to the measurement data in the entire slenderness range of the tested paper samples.

## 2. Load during the Destruction Phase of the Paper Web

Determining the load capacity of a compressed paper web requires an analysis of its behaviour in the conditions of buckling. This phenomenon has been extensively studied by Kołakowski et al., Zaczynska et al. and Kubiak et al. [35,36,37] in relation to various materials.

Equations (1)–(7) define the formulas for the destructive load of the compressed paper sample, which were verified by experimental tests. In the strength of materials, the two most commonly used calculation methods are the method of ultimate stress (i.e., strength method) and the method of limit load (i.e., method of state). The second method, i.e., the limit load capacity method, was used in the study. In the limit load capacity method, a model of a perfectly rigid plastic body is adopted, in which the limits of proportionality, elasticity, plasticity and strength are the same and correspond to the yield strength *Re*. Królak and Murray et al. [38,39] proposed the use of two true plastic mechanisms for the destruction of the paper web: mechanism 1—a one-hinge mechanism in the plate, and mechanism 2—a three-hinge mechanism in the plate.

### 2.1. Mechanism 1 of Destruction

The shape of a paper sample subjected to compressive forces is shown in Figure 1a. More figures showing the different stages of paper compression can be found in [23]. According to Królak [38] and Murray, Khoo [39], mechanism 1 of the destruction of the paper web with a thickness *g*, length *l* and width *b*, freely supported on both loaded edges has the form presented in Figure 1b. The length *l* is not the length of the entire sample but rather the distance between the holders of the UTM, referred further to as the fastening length.

The maximum load during the destruction phase of the paper web simply supported at both ends can be assumed as described in Equation (1) [38,39]:(1)$$P_{1}=R_{e}bg\left[\sqrt[{2}]{{\left(\frac{2\delta }{g}\right)}^{2}+1}-\frac{2\delta }{g}\right]$$
where in destruction mechanics it is assumed that the compressive strength and yield strength are identical. For the purpose of developing yield strength, $R_{e}$ is determined from the SCT test for a standard fastening length of 0.7 mm. For a perfectly rigidly plastic model, it can be assumed that *SCT* [N/m] = *R_e_* [Pa] *· g* [m]. Hence, $R_{e}=\frac{SCT}{g}$. δ denotes the deflection arrow of the web.

Based on Equation (1), the shortening of the web can be determined (Figure 2).

The paper web with length *l* is shortened due to load, which can be written as described in Equation (2).
(2)$$∆l=l-2AB=l\left(1-\sqrt[{2}]{1-{\left(\frac{2\delta }{l}\right)}^{2}}\right)$$

On the other hand, the shortening of the web can be written from Hooke’s law:(3)$$P_{1}=E_{MD}bg\frac{∆l}{l}$$

By substituting (2) to (3), the value of the load can be determined as a function of the deflection arrow *δ*. The intersection of the curve (1) with the line (3) in the coordinate axis system *P*_1_ (∆*l*) allows us to determine the destructive load of the web as a function of shortening. This method of determining the destructive load will be further referred to in the article as the M1 model.

### 2.2. Mechanism 2 of Destruction

According to Królak [38], and Murray, Khoo [39], mechanism 2 of the destruction of the web fixed at both loaded ends has the form shown in Figure 3.

The maximum load in destruction phase 2 can be assumed as follows [38,39]:(4)$$P_{2}=R_{e}bg\left[\sqrt[{2}]{{\left(\frac{\delta }{g}\right)}^{2}+1}-\frac{\delta }{g}\right]$$

Details of the shape geometry of the sample fixed at the edges are shown in Figure 4.

Taking into account the results of research presented in [23], two cases of shortening the paper web of length *l* were considered:
Case 2A: AB = BC = CE = EF = *l*/4 (Figure 4):
(5)$$∆l=l-l/2-2\frac{l}{4}\sqrt[{2}]{1-{\left(\frac{4\delta }{l}\right)}^{2}}=l/2\left(1-\sqrt[{2}]{1-{\left(\frac{4\delta }{l}\right)}^{2}}\right)$$Case 2B: AB = EF = 3*l*/8; BC = CE = *l*/8 (Figure 4):
(6)$$∆l=l-3l/4-2\frac{l}{8}\sqrt[{2}]{1-{\left(\frac{8\delta }{l}\right)}^{2}}=l/4\left(1-\sqrt[{2}]{1-{\left(\frac{8\delta }{l}\right)}^{2}}\right)$$

As for mechanism 1, the shortening of the web can also be written from Hooke’s law:(7)$$P_{2}=E_{MD}bg\frac{∆l}{l}$$

The point of intersection of the curve (4) with line (7), taking into account (5) in the coordinate system *P*_2*A*_ (∆*l*), allows us to determine the destructive load of the paper web fixed at the loaded ends for *l*/4. This method of determining the destructive load will be referred to later in this article as the M2A model.

However, the point of intersection of the curve (4) with the line (7) for (6) allows us to determine the relationship *P*_2*B*_ (∆*l*) for *l*/8. This method of determining the destructive load will be referred to later in this article as the M2B model.

The authors do not know of examples in the world literature about using true plastic mechanisms to determine the load capacity of paper webs. The results for steel plates are included, among others, in work by [38,39,40,41].

### 2.3. Validation of Destructive Load Formulas for Mechanism of Destruction 1 and 2

The SCT test allows us to determine the value of the destructive load, which, according to the perfectly rigid plastic model, corresponds to the yield strength and at the same time the compressive strength, i.e., *R_e_*. The analysis of Equations (1) and (4) leads to the conclusion that in the SCT sample, the maximum force values obtained from these equations are lower than those determined experimentally. This is because the expressions in square brackets in Equations (1) and (4) are less than one. According to the results of work from [23], a paper web with a standard fastening length *l* = 0.7 mm behaves similarly to the M2B destruction mechanism for length *l*/8, i.e., according to (6). This allows the validated value of *R_e_* to be determined according to Equations (4), (6) and (7) for *l*/8. In the further part of the work, only these validated values of *R_e_* are given, so that the SCT values for M2B determined from these formulas correspond to the experimental SCT values.

## 3. Measurement Results

The research included tests of crushing webs of packaging paper in a universal testing machine (UTM) Zwick Roell Z 010 equipped with precise hydraulically clamped handles. Thanks to this, it was possible to maintain the same clamping force in each test and to precisely set the value of zero preload force. The UTM used for the measurements allowed only the testing of paper webs with a maximum length of 5.0 mm. During the tests, a series of images of the edges of the compressed paper was made, which allowed us to observe changes in its shape and experimentally determine the size of the deflection arrow of the sample. Detailed results are presented in [23].

Five packaging papers marked with symbols from Pa1 to Pa5 were selected for the tests, differing in weight, thickness, value of Young’s modulus and the value of destructive force in the SCT test. The papers were previously air-conditioned in an atmosphere with a temperature of 23 °C and a humidity of 50% with a standard procedure for pre-drying the samples to avoid inaccuracies due to the occurrence of hysteresis depending on the moisture content of the paper on the relative humidity of the air [42].

The material parameters of individual papers are summarized in Table 1.

In all tests, the width of the samples, as in the SCT test, was equal to *b* = 15 mm. Tests of each paper were performed for the following fastening lengths: *l* = 0.7 mm, 1.3 mm, 2.0 mm, 2.5 mm, 3.0 mm, 3.5 mm, 4.0 mm, 4.5 mm and 5.0 mm. For each fastening length, 10 measurements were made and the results were averaged. The obtained values of the maximum force are presented in Table 2. Due to the low practical significance of a web with very high slenderness, at which the load capacity decreases significantly, the tests were limited to slenderness of no more than 140.

## 4. Discussion of Research Results

The maximum force endured by the paper sample depends on the slenderness of the sample, $\frac{l}{i}$ where *l* is the length of the web fastening in the UTM holders and *i* is the gyration radius. Therefore, changes in the maximum forces of all papers were analysed as a function of slenderness *s*, defined as:(8)$$s=\frac{l}{i}=\frac{l}{\sqrt{\frac{I_{min}}{A}}}=\frac{l}{\sqrt{\frac{bg^{3}}{12bg}}}=\frac{l\sqrt{12}}{g}\approx \frac{3.5l}{g}$$

In the theory of thin plates [38,43,44,45,46] the classification of thin-walled structures is assumed as follows:
When *s* > 25–30 we have thin-walled plates;When 5 ≤ *s* < 25 we have plates of medium thickness;When *s* < 5 we have thick plates [23,44].

The most commonly used theory of medium-thickness plates is called the first-order shear deformation theory (FSDT) [23,43,46]. The formulas used in the work were derived from the theory of thin plates [38,39,40,41,42,43,44,45].

The article [23] presents pictures of the paper web at the time of destruction for 5 different clamping lengths. For a length of 0.7 mm, the paper fibres were delaminated and crushed near the fixed, lower UTM handle. In this case, the slenderness of the sample is *s* = 5. For lengths of 1.3 and 2.0 mm, destruction also occurs near the lower handle with visible horizontal displacement of the samples. For these cases, we have *s* = 9.3 and *s* = 14.3, respectively. For lengths 3.0 (*s* = 21.4) and 4.0 mm (*s* = 28.5), the destruction of the specimens occurs in the central part and the sample becomes V-shaped, so the webs enter tangentially into the clamping surfaces of both machine holders. However, for a length of 5 mm (*s* = 35.7), the V-shape is very clearly visible, especially the kink. The sample in both jaws practically behaves as for the simply supported.

## 5. Predicting the Maximum Compressive Force of a Paper Sample

After testing all paper samples, the accuracy and applicability of the models described in Chapter 2 were assessed in a similar way as in Pyryev et al. [47]. Table A1, Table A2, Table A3, Table A4 and Table A5 in Appendix A show the results of measurements and predictions of the maximum compressive force that the tested papers can withstand. M1, M2A and M2B models were used for calculations.

As a result of the accuracy analysis of the maximum load capacity prediction with individual models, it was concluded that the M2B model did not describe the actual measurement data well and the further focus was on comparing the other two models. The results of predicting the maximum force acting on the test sample using models M1 and M2A are shown in Figure 5, Figure 6, Figure 7, Figure 8 and Figure 9.

In all cases, the M2A model performs better for samples with low slenderness. As the slenderness of the samples increases, the M1 model begins to outperform the M2A model in terms of accuracy. Using a single model to predict the maximum compressive force over the entire slenderness range does not seem to be the correct approach. As a result of further research, a model was proposed that combines the M1 model used for larger slenderness and the M2A model used for smaller slenderness. It is not possible to find the limit of slenderness at which to go from M2A to M1 by comparing the dependence of the maximum force on the slenderness, as it depends on the type of paper, as can be seen in Figure 10.

In order to compare the accuracy of the models applied to different papers, relative errors in the prediction of the maximum force were determined in each case. These errors were defined as the difference between the measured and calculated value divided by the measured value. It was observed that in most cases the M2A model provided better predictions for low slenderness, while the M1 model performed better for larger slenderness values. To establish the limit applicability of both models, the model error resulting from using M1 for slenderness values above a certain threshold *sg*, and M2A for slenderness values below *sg* was defined as the root mean square (*RMS*) value of errors specific to each model within the corresponding slenderness range:(9)$$RMS\left(sg\right)=\sqrt{\frac{\sum_{s<sg}{\left(\frac{F_{max}-P2A}{F_{max}}\right)}^{2}+\sum_{s\geq sg}{\left(\frac{F_{max}-P1}{F_{max}}\right)}^{2}}{N}}\cdot 100,\%$$
where *N* is the number of relative errors determined.

The determined dependence of the RMS modelling error on the limit slenderness sg value is shown in Figure 11. The smallest error value was obtained in the slenderness range from 92 to 95. Finally, *sg* = 93.5 was adopted as the limit value for the applicability of the M1 and M2A models.

The relative errors of the maximum compressive force prediction were then estimated using M1 and M2A models. The dependence of the error on the slenderness of the sample for individual papers is shown in Figure 12, Figure 13, Figure 14, Figure 15 and Figure 16. In the case shown in Figure 16, the maximum slenderness value is lower than the assumed value of *sg* = 93.5 and, therefore, only the M2A model is applicable.

The obtained relative error values for predicting the maximum compressive force of the sample are within the acceptable range for heterogeneous materials from −30% to 30%. In the case of the Pa5 paper, due to its large thickness, the slenderness range was limited to 86.2, which meant that the M2A model applies to the entire range of slenderness tested for this paper.

The RMS error in predicting the load capacity of individual papers using a combination of the M1 and M2A models is shown in Table 3.

## 6. Development of a Model for Calculating the Maximum Compressive Force Based on the Empirical Formula

Figure 10 shows the relationships of the maximum forces withstood by individual papers at different slenderness. Because of variations in fibrous composition, weight, and thickness, the maximum force measurements for samples of different papers with the same slenderness do not align. To avoid these discrepancies, the force was related to the value of the maximum forces obtained for individual securities with a short fastening length of 0.7 mm.

As a result, the relative value *F_rel_* of the maximum force was defined as the ratio of the measured maximum compressive force *F_max_* to the maximum compressive force with a fastening length of 0.7 mm *F_max_*_0.7_ and expressed in %:(10)$$F_{rel}=\frac{F_{max}}{F_{max0.7}}\cdot 100,\left[\%\right]$$

The results of the application of Equation (10) to all examined papers are shown in Figure 17.

As can be seen, the individual graphs are similar to each other and can be described with high accuracy by one dependence. Figure 18 shows an approximation of all results with a polynomial of third degree. Further, 1st-degree and 2nd-degree polynomial approximations were also studied. The results are presented in Appendix B. The approximation errors were larger in both cases. The use of polynomials of higher degrees for the approximation of measurement data is not advisable due to the unjustified complication of the model without a noticeable reduction of approximation errors. Using 3rd degree polynomial the approximation of the relative maximum compressive force *F_rel_* is described by the following equation:(11)$${\tilde{F}}_{rel}=4\cdot 10^{-5}s^{3}-0.0089s^{2}-0.0851s+103.46$$

The relative prediction errors *F_rel_* as a function of slenderness for approximation by a third-degree polynomial are shown in Figure 19.

Given Equation (11) and the value of the maximum compressive force of the sample *F_max_*_0.7_, with a fastening length of 0.7 mm, the value of the maximum force *F_max_* can be calculated, with different slenderness s of the paper samples:(12)$$F_{max}=F_{max0.7}\cdot {\tilde{F}}_{rel}$$

## 7. Conclusions

Analytical and empirical models were used to assess the maximum forces carried by the compressed paper by the forces acting in its plane. None of the analytical models presented in the paper was able to describe changes in maximum force as a function of the slenderness of samples in the entire analysed range. One of the analytical models, due to worse accuracy, was rejected. For the other two models (M1 and M2A), the limit slenderness *sg* = 93.5 determining the slenderness ranges in which they apply, was determined.

The M2A model is used for a paper slenderness value of 93.5, and the M1 model is used above this slenderness. For Pa5 paper, whose slenderness did not exceed 86.2, only M2A model was used. In this case, the *RMS* approximation error is about 7%. In other cases, both models were used. The *RMS* error values did not exceed 25.1%.

Given the empirical Equation (12) and the value of the force measured in the paper SCT, the value of the maximum force *F_max_* can be calculated, with different slenderness *s* of the paper samples. The *RMS* error for calculating the maximum compressive forces on the basis of the empirical relationship in the examined cases presented in Figure 19, is 7.0% which makes the empirical model the most suitable for analysing the compression of packaging papers.

The results of the tests and the development of a model describing the maximum compressive force acting on the paper in its plane will be used to describe the strength of corrugated board, paper cores and other paper products.

## Figures and Tables

**Figure 1 materials-16-04544-f001:**
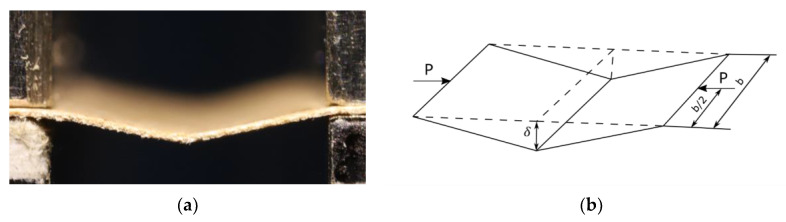
(**a**) Side view of compressed paper sample, (**b**) mechanism 1 of paper web destruction.

**Figure 2 materials-16-04544-f002:**
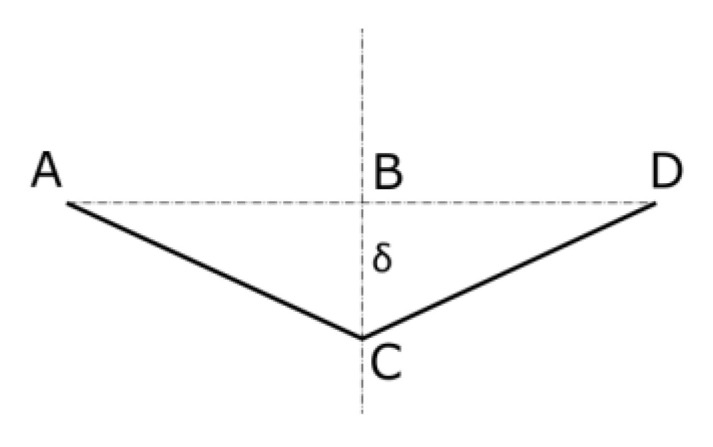
The geometry of the mechanism of the destruction of the paper web.

**Figure 3 materials-16-04544-f003:**
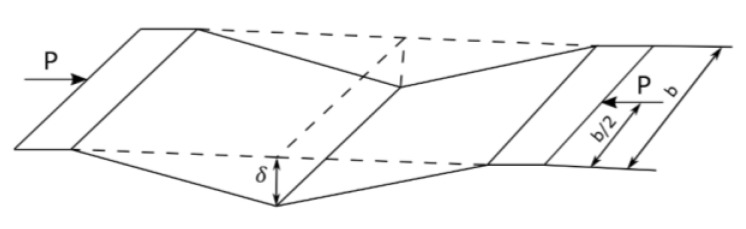
The geometry of the mechanism of destruction of the paper web fixed at both ends.

**Figure 4 materials-16-04544-f004:**
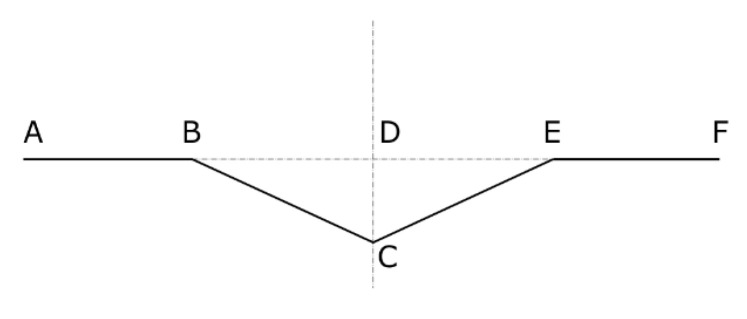
Shortening the paper web for mechanism 2 of destruction.

**Figure 5 materials-16-04544-f005:**
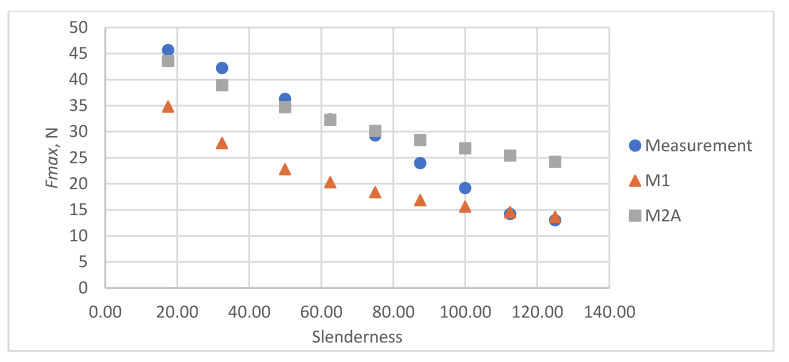
Results of measurements and calculations of the maximum compressive forces a function of the slenderness of Pa1 paper sample.

**Figure 6 materials-16-04544-f006:**
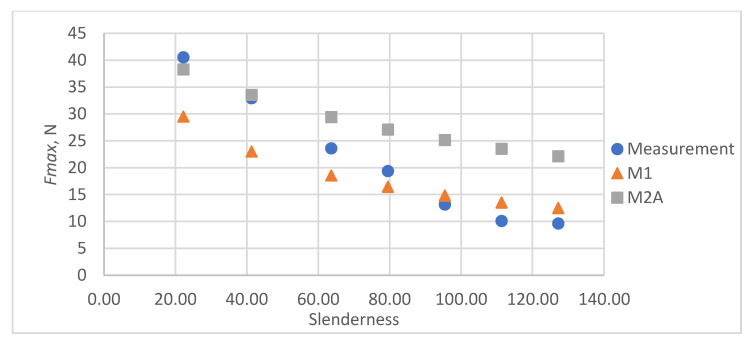
Results of measurements and calculations of the maximum compressive forces a function of the slenderness of Pa2 paper sample.

**Figure 7 materials-16-04544-f007:**
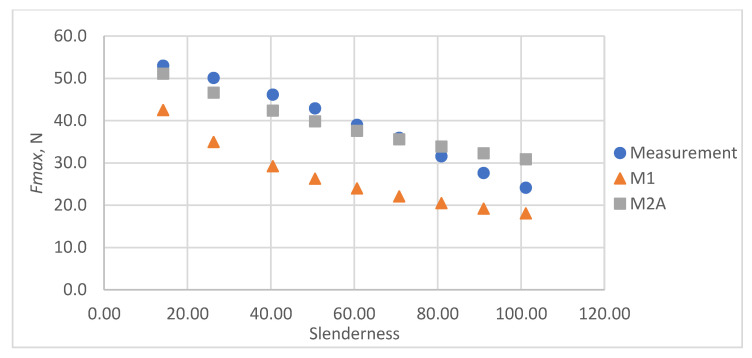
Results of measurements and calculations of the maximum compressive forces a function of the slenderness of Pa3 paper sample.

**Figure 8 materials-16-04544-f008:**
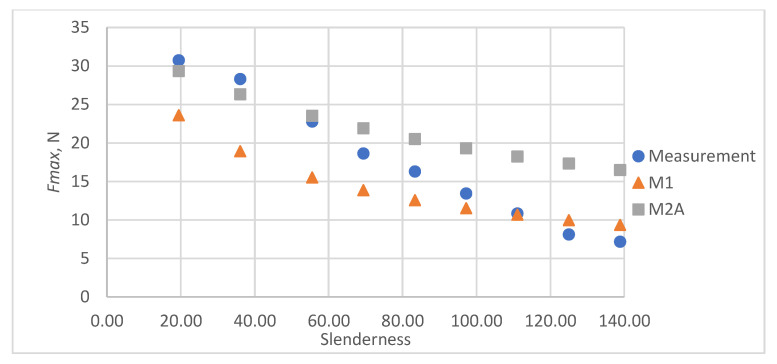
Results of measurements and calculations of the maximum compressive forces a function of the slenderness of Pa4 paper sample.

**Figure 9 materials-16-04544-f009:**
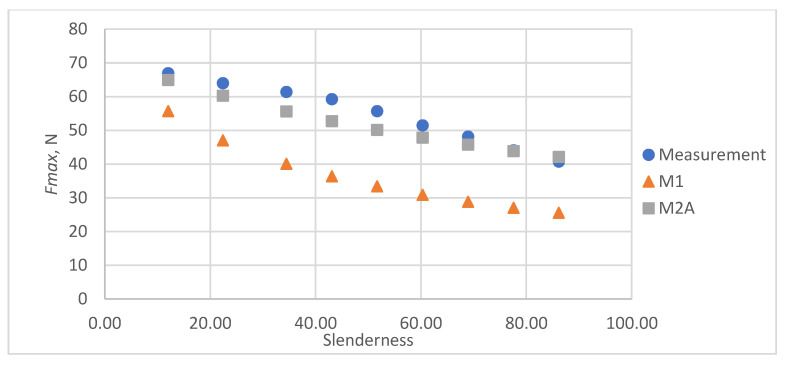
Results of measurements and calculations of the maximum compressive forces a function of the slenderness of Pa5 paper sample.

**Figure 10 materials-16-04544-f010:**
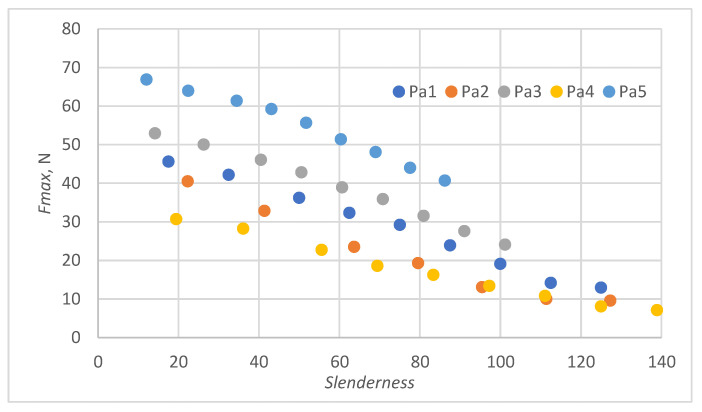
Dependence of the maximum compressive force of the paper sample on slenderness.

**Figure 11 materials-16-04544-f011:**
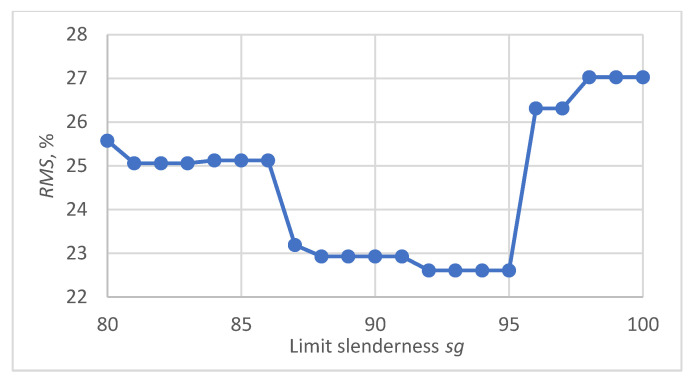
Dependence of the *RMS* error of predicting the maximum compressive force using the model resulting from the assembly of M1 and M2A on the limit value of slenderness.

**Figure 12 materials-16-04544-f012:**
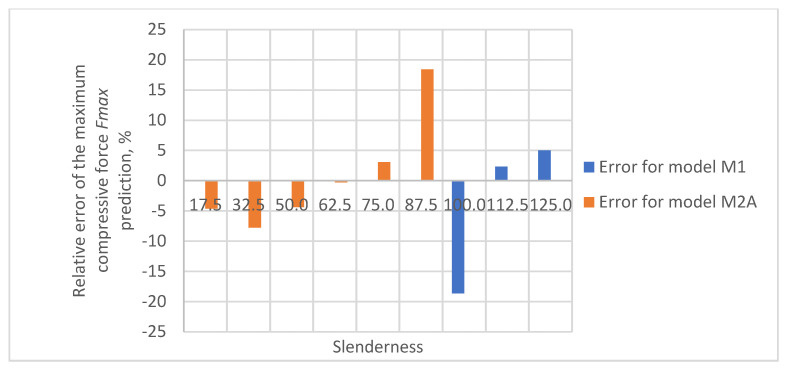
Relative error of predicting maximum compressive force as a function of slenderness of Pa1 paper sample using models M1 and M2A.

**Figure 13 materials-16-04544-f013:**
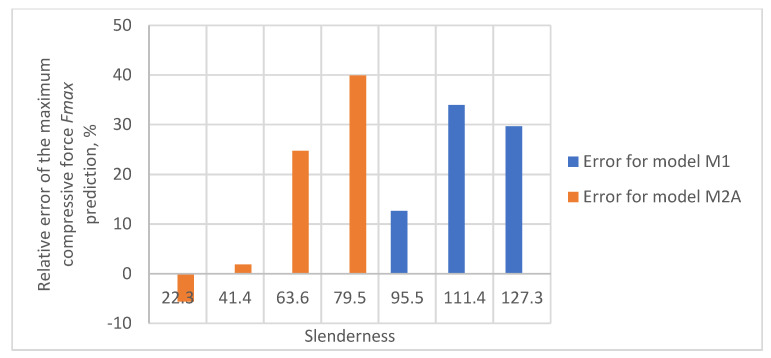
Relative error of predicting maximum compressive force as a function of slenderness of Pa2 paper sample using models M1 and M2A.

**Figure 14 materials-16-04544-f014:**
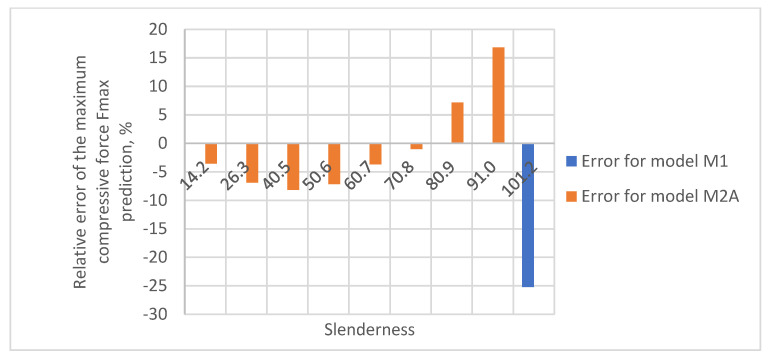
Relative error of predicting maximum compressive force as a function of slenderness of Pa3 paper sample using models M1 and M2A.

**Figure 15 materials-16-04544-f015:**
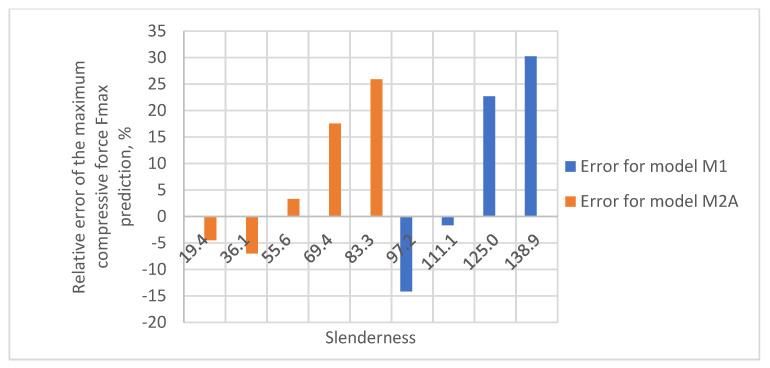
Relative error of predicting maximum compressive force as a function of slenderness of Pa4 paper sample using models M1 and M2A.

**Figure 16 materials-16-04544-f016:**
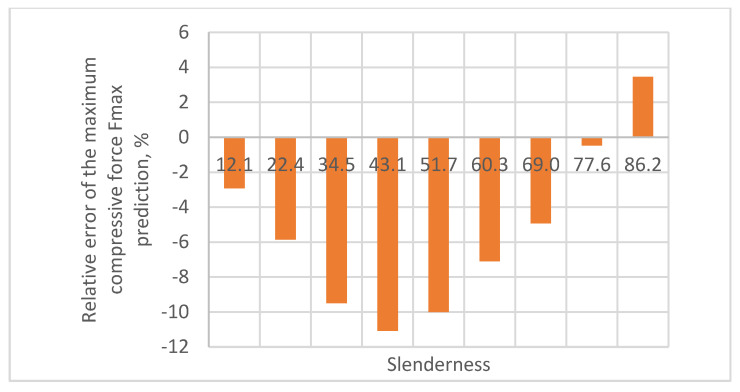
Relative error of predicting maximum compressive force as a function of slenderness of Pa5 paper sample using model M2A.

**Figure 17 materials-16-04544-f017:**
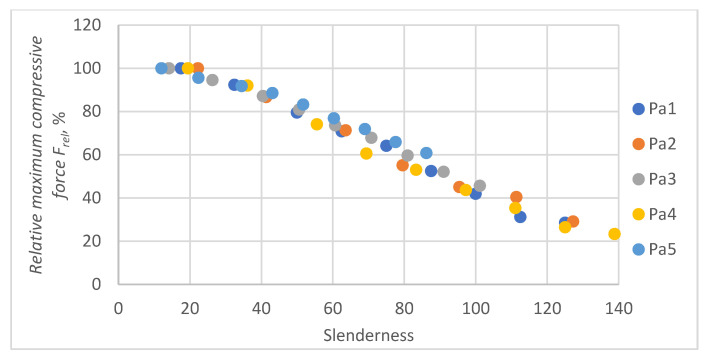
Value of the relative maximum compressive force *F_rel_* as a function of slenderness of the sample.

**Figure 18 materials-16-04544-f018:**
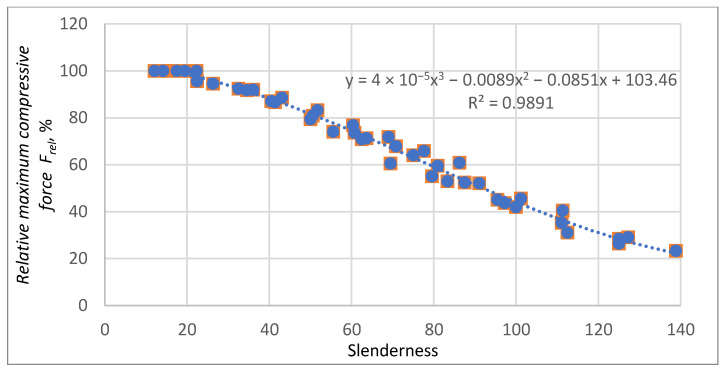
Prediction of the relative maximum compressive force *F_rel_* as a function of the slenderness of the sample by third-degree polynomial.

**Figure 19 materials-16-04544-f019:**
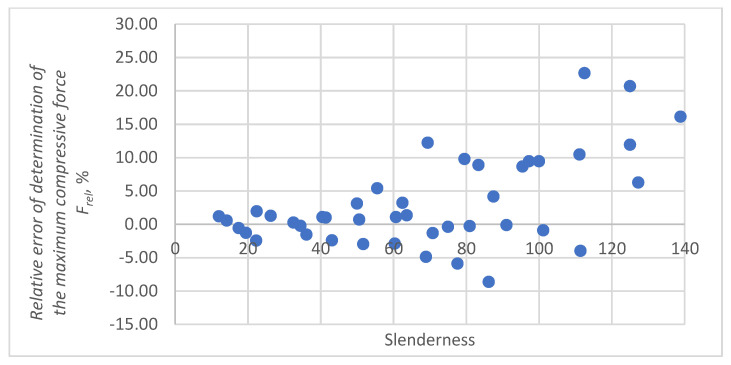
Relative errors of prediction a relative maximum compressive force *F_rel_* by a third-degree polynomial.

**Table 1 materials-16-04544-t001:** Average values of material parameters of the tested papers and their standard deviation.

Paper	Young’s Modulus in MD *E*, GPa	STD of Young’s Modulus, GPa	Thickness *g*, mm	STD of Thickness, mm	Yield Strength *Re*, MPa	STD of Yield Strength, MPa
Pa1	5.61	0.18	0.140	0.01	21.8	1.40
Pa2	6.92	0.10	0.110	0.007	18.1	0.85
Pa3	5.63	0.12	0.173	0.01	20.4	1.02
Pa4	5.52	0.10	0.126	0.005	16.2	0.73
Pa5	6.81	0.19	0.203	0.008	22.0	0.87

**Table 2 materials-16-04544-t002:** Average values and standard deviation of the maximum compressive force of the test samples of the papers.

Length of Fastening *l*, mm	Average Value of *F_max_*, N					Standard Deviation of *F_max_*, N				
	Paper					Paper				
	Pa1	Pa2	Pa3	Pa4	Pa5	Pa1	Pa2	Pa3	Pa4	Pa5
0.7	45.7	29.8	53.0	30.7	66.9	2.95	1.41	2.65	1.38	2.64
1.3	42.2	25.9	50.1	28.3	64.0	2.81	1.80	2.00	1.33	1.86
2	36.3	21.3	46.2	22.8	61.4	3.33	1.05	2.23	1.63	2.26
2.5	32.4	16.4	42.9	18.6	59.3	2.90	0.85	1.64	1.24	2.45
3	29.3	13.5	39.0	16.3	55.7	2.57	0.76	1.07	0.96	1.13
3.5	24.0	12.1	36.0	13.4	51.5	2.44	0.79	1.78	0.49	3.22
4	19.2	8.7	31.5	10.9	48.1	1.95	0.52	1.41	0.38	2.48
4.5	14.2	6.9	27.6	8.1	44.1	1.67	0.73	1.33	0.57	2.48
5.0	13.0	5.2	24.2	7.2	40.7	1.61	0.59	1.08	0.42	1.79

**Table 3 materials-16-04544-t003:** *RMS* value of relative load capacity prediction error using a combination of M1 and M2A.

Paper	Pa1	Pa2	Pa3	Pa4	Pa5
*RMS*, %	9.57	25.15	11.37	17.28	7.01

## Data Availability

The data presented in this study are available on request from the corresponding author. The data are not publicly available due to the high degree of complexity of their organization. The authors have not yet developed an appropriate standard for their storage.

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
