# Peer review of "New Models for Calculating the Maximum Compressive Force of Paper in Its Plane"

_materials, 2023, doi:10.3390/ma16134544_

Round 1

Reviewer 1 Report

Sl No

Description

Recommendation

1

Abstract

Rearrange the structure of abstract

At first sentence objective of this work will make the abstract more attractive

2

Introduction

For the general reader please make clear why this work is taken.

What is the research gap? please make clear with literature support

What is linear layer, flute layer and fastening length?

If possible, please give detail of the literature when add citation in introduction section.

End of the introduction a guideline of the manuscript will be helpful for the reader.

Experimental procedure

3

Line 60

Please mention the name of mechanisms.

4

Fig 1 and Fig 2 approximately same except notation A, B and C. If both Figures are necessary, please make them significantly different.

In both Fig please identify the thickness and length of the paper.

SCT= ? give detail when use any abbreviation 1st time in the manuscript

5

Line 64

Please remove  the 2nd “on” in the sentence

6

Line 72

Please check the fastening length value 0.7

7

Line 99

In this study two cases are considered… please give support of your consideration.

Please be constant when writing notation in the manuscript

e.g. g, l will  be italic everywhere

8

Line 121

2B will be  M2B

9

Please include photographs of raw materials/ paper, which will give a clear picture of input for this work in the methodology section.

Measurement Results

10

Please include the findings of other authors( related to this manuscript ) during addition of citation

11

Line 127

Please clear the sentence.

12

Line 131

How many samples were tested in the previous work?

13

Line 135

From the previous work how 5 samples were selected for this study?

14

Table 1

“g” and “Re” will be in italic

Table 2

”l“ will be italic

15

Line 164-166

Please recheck the assumptions regarding thick and thin plates

16

Line 177

Please capitalize the first letter of the sentence.

Please give literature support of your findings of section 5

17

Line 290

“….this paper” -- makes confusion.

Can be replaced by “M2A model applies to the entire range of slenderness for the tested 5 papers”

18

Line299- 300

Please make the sentence simple

Fig 19

Please include R2 value in the figure

19

Please include recommendations for further development.

Conclusion

20

Line 342

How do you get 7% please give support of this conclusion in the result section. From which Fig please mentioned in the results section.

21

If possible have it revised by native speaker. Please check spellings, grammar and content again before final submission.

Sl No

Description

Recommendation

1

Abstract

Rearrange the structure of abstract

At first sentence objective of this work will make the abstract more attractive

2

Introduction

For the general reader please make clear why this work is taken.

What is the research gap? please make clear with literature support

What is linear layer, flute layer and fastening length?

If possible, please give detail of the literature when add citation in introduction section.

End of the introduction a guideline of the manuscript will be helpful for the reader.

Experimental procedure

3

Line 60

Please mention the name of mechanisms.

4

Fig 1 and Fig 2 approximately same except notation A, B and C. If both Figures are necessary, please make them significantly different.

In both Fig please identify the thickness and length of the paper.

SCT= ? give detail when use any abbreviation 1st time in the manuscript

5

Line 64

Please remove  the 2nd “on” in the sentence

6

Line 72

Please check the fastening length value 0.7

7

Line 99

In this study two cases are considered… please give support of your consideration.

Please be constant when writing notation in the manuscript

e.g. g, l will  be italic everywhere

8

Line 121

2B will be  M2B

9

Please include photographs of raw materials/ paper, which will give a clear picture of input for this work in the methodology section.

Measurement Results

10

Please include the findings of other authors( related to this manuscript ) during addition of citation

11

Line 127

Please clear the sentence.

12

Line 131

How many samples were tested in the previous work?

13

Line 135

From the previous work how 5 samples were selected for this study?

14

Table 1

“g” and “Re” will be in italic

Table 2

”l“ will be italic

15

Line 164-166

Please recheck the assumptions regarding thick and thin plates

16

Line 177

Please capitalize the first letter of the sentence.

Please give literature support of your findings of section 5

17

Line 290

“….this paper” -- makes confusion.

Can be replaced by “M2A model applies to the entire range of slenderness for the tested 5 papers”

18

Line299- 300

Please make the sentence simple

Fig 19

Please include R2 value in the figure

19

Please include recommendations for further development.

Conclusion

20

Line 342

How do you get 7% please give support of this conclusion in the result section. From which Fig please mentioned in the results section.

21

If possible have it revised by native speaker. Please check spellings, grammar and content again before final submission.

Author Response

The authors would like to thank the Reviewer for valuable comments. We fully agree with the Reviewer.

Reviewer 3 Report

overall, there is no major issue. my only suggestion is regarding the short-span compression test. it first appears in subsection 2.1, I would suggest authors write it in full name for the first time with the abbreviation in the parentheses, then the rest can remain is SCT. this will help the reader to understand better.

short-span compression test(SCT)

there are some small suggestions regarding the English,

for example, in the abstract, authors can directly just write "this article......", instead of "the article". 

Please confirm with the editor, commonly for the table/figure citation, we don't need to write it in the plural. for example, Table 3-7, instead of Tables 3-7.

Author Response

The authors would like to thank the Reviewer for valuable comments. Corrections were made according to Reviewer's comments.

Reviewer 4 Report

The presented article is a continuation of the authors' work published earlier.  Some remarks are as follows:

- l53 - note formula's numbers instead of mentioning bellow
- l159 - write variables denominations for formula (8)
- presentation of results - results of tables are then presented in graphs, do not double presentation of results
- l251 - authors introduce limit applicability sg which is denominated in Fig 11 as limit slenderness,but it is not obvious or explained how this number is determined and why the range from 80 to 105 is chosen - this section must be significantly improved
- further results (Figs 12 - 16) are direct consequence of sg (I suppose) and there is no explanation on usage of models. In Fig 16 only M2A model is applied, on other papers, both models are applied etc. please give more insight on sg, i.e. model, choice
- the authors introduce equation 10 and present the results  in Fig 17, but I can't understand from which results the polynomial appproximation was set and does this approximation take into account SD of measurements or similar? Do the authors have any insight and theroetical knowledge that this dependance should be polynomial of third degree?
- equation 12 looks to come from the equation 10, but they are not mathematically correct. If the Frel in (12) is different to Frel in (10), it should be stressed by different denomination.

The authors should significantly improve the presentation of results and give more explanation of some calculations which are made to give more scientific relevance. It would not be significant results if the results and approximations are applicable only on the 5 tested papers!

There are some typos and the language should be more smooth as in this form it is hard to read.

Reviewer 5 Report

This article is well written, comprehensive and logically organized, and contains valuable information on the new models for calculating the maximum compressive force of paper in its plane. The authors did excellent research on investigating the analytical and empirical models used to assess the maximum forces carried by the compressed paper by the forces acting in its plane. The authors demonstrated the performed tests are directly applicable in the calculation of the mechanical properties of corrugated cardboard and the determination of the load capacity of cardboard packaging. This manuscript does not contain much error analysis on the material parameters of the tested papers which is highly required for readability purposes. The authors presented the material parameters of the tested papers in Table 1. It is suggested the authors should place the standard deviations of Young's modulus, Thickness, and Yield strength for the reliability and readability of the present research. The submitted manuscript has significant scientific insights and the conclusions are soundly supported by the experimental data. However, the present submission requires minor revisions before being considered for publication in the well-circulated Materials in its current condition.

Abstract: The article presents the results of research on crushing paper with compressive forces acting in its plane. Samples of various packaging papers with different fastening lengths were examined. It was shown that due to the specific structure of the paper and the high heterogeneity of its structure, packaging paper is a material difficult to determine the maximum compressive stress. Three analytical models describing the load capacity of a flat paper web were investigated and an alternative empirical model was proposed. The results of the performed tests are directly applicable in the calculation of the mechanical properties of corrugated cardboard and the determination of the load capacity of cardboard packaging.

Author Response

The authors would like to thank the Reviewer for valuable comments.. We fully agree with the Reviewer. Standard deviations have been added to the table 1. The text was also extensively revised.

Reviewer 6 Report

This paper does not have the structure of a scientific paper. Data concerning theory, experiments, and results are mixed. This results in a lack of consistency and sequence, as well as a difficult summarization of the content. To determine the mechanical properties of the paper, the technical specifications of the cardboard that were used, the standards by which the measurements were made, as well as the devices, are missing, which makes it impossible to repeat the experiments and verifiability.

The English in the manuscript needs a substantial improvement. In addition to grammatical errors, the syntax is also needed.

The introduction of the study is short with a lot of references cited. This paragraph should be slightly rewritten and expanded. It is not usual to cite 34 references in 6 sentences. Thus, bulk references should be avoided.

The second paragraph should be a part of the introduction.

Line 53: The common scientific language is not ‘formula’ but “equation”. Please, check the whole manuscript.

Line 63: According to [38,39] – According to name et al. and name et al. [38,39]. Please, check the whole manuscript.

Line 69...in the form. – as described in Eq. 1. Please, check the whole manuscript.

Line 72: what does SCT mean?

Line 99: Should not the l in length l be written in italics?

Line 126: not measurement results, but results

Line 153: Discussion not analysis of research results

Lines 182 – 209: how were the conditions determined?

the work is written in slang and with the absence of professional scientific dictionary

Round 2

Reviewer 4 Report

Dear Authors,

the paper is improved, but there are two points which I must stress, again.

- first it is not common to duplicate presentation of results. If it is necessary, I would suggest giving tabular presentation as additional materials. In this presentation, the paper is longer without giving additional information.

- second, the explanation about using the 3rd degree polynomial approximation is that the other two give higher error. As there are no errors in measurements presented, no verification of the equation (11) by testing some other papers, it is hard to firmly claim that the presented equation is correct and widely usable for Fmax calculations.

Some minor improvents are possible, but not significant.

Reviewer 5 Report

Dear Authors: Many thanks for all of your sincere efforts in improving your manuscript. From my point of view, the revised article is highly satisfactory, and therefore, merits acceptance for publication in the Materials.

Author Response

Dear Reviewer,

we would like to thank You for your positive opinion.

Kind regards
authors

Reviewer 6 Report

The changes made are only cosmetic. The paper still lacks in scientific contribution.

The part from lines 161 -174 is an experimental part

How did the authors include the paper properties in the modles calculation (weight, structure, the influence of atmosferic conditions in which the paper is stored). How can others materials calculations be relevant for paper properties determination and predicitions through the models made?

line 201- refrence 444

in table 1- The usual presentation of the results is mean +- STD. there is no need for additional column. How can thickness be in g and mm?

style and grammatical errors
